# CAT: Content-Adaptive Image Tokenization

## Abstract

Most existing image tokenizers encode images into a fixed number of tokens or patches, overlooking the inherent variability in image complexity and introducing unnecessary compute overhead for simpler images. To address this, we propose **C**ontent-**A**daptive **T**okenizer (**CAT**), which dynamically adjusts representation capacity based on the image content and encodes simpler images into fewer tokens. We design (1) a *caption-based evaluation system* that leverages LLMs to predict content complexity and determine the optimal compression ratio for an image, and (2) a novel *nested VAE architecture* that performs variable-rate compression in a single model. Trained on images with varying complexity, CAT achieves an average of **15% reduction in rFID** across **seven** detail-rich datasets containing text, humans, and complex textures. On natural image datasets like ImageNet and COCO, it **reduces token usage by 18%** while maintaining high-fidelity reconstructions. We further evaluate CAT on two downstream tasks. For image classification, CAT consistently improves top-1 accuracy across **five** datasets spanning diverse domains. For image generation, it **boosts training throughput by 23%** on ImageNet, leading to more efficient learning and improved FIDs over fixed-token baselines.

## 1. Introduction

Image tokenizers compress high-resolution images into low-dimensional latent representations, enabling compact and semantically meaningful inputs for downstream tasks such as generation and classification (Esser et al., 2020; Kingma and Welling, 2014; Yu et al., 2024a; Shen et al., 2022; Tu et al., 2022; Yuan et al., 2021a; Mentzer et al., 2023; Yu et al., 2024b). Despite their effectiveness, most existing tokenizers operate at a fixed compression ratio, producing latent representations of uniform length regardless of the image's content. However, natural images exhibit significant variability in complexity, from sparse scenes to densely textured ones, suggesting that fixed-length representations can be both inefficient and suboptimal.

Classical codecs such as JPEG (Wallace, 1992) implicitly exploit this variation: when fixing the quality level, they produce different file sizes for images with different frequency characteristics. In contrast, fixed-ratio tokenizers may under-compress simple images, wasting compute on redundant information, or over-compressing complex ones, losing important details. These problems become more prominent when tokenizers are used in large-scale generative pipelines or as feature extractors in downstream tasks.

Several recent works explore dynamic token representation during inference (Yan et al., 2024; Duggal et al., 2024). However, these methods typically require access to the *input image*—an assumption incompatible with many practical use cases. For example, in image generation with latent diffusion models (Rombach et al., 2021), only the user's text prompt is available, and the number of latent tokens, which significantly influences generation quality, must be specified in advance. Moreover, these methods do not adapt tokenizer *training* to image complexity, missing an opportunity to optimize for both content and downstream utility.

In this work, we introduce **Content-Adaptive Tokenizer (CAT)**, a novel approach that dynamically adjusts representation capacity based on image complexity. CAT combines a *caption-driven complexity evaluator* with a *nested autoencoder architecture* to produce variable-length latent features in a single forward pass (Figure 1). Specifically, the evaluator uses large language models (LLMs) to predict the optimal compression ratio from textual descriptions. It analyzes the image's caption and answers perception-oriented questions (e.g., *"are there human faces or text?"*) to produce an interpretable complexity score. Based on this score, we assign one of three compression ratios to the image: 8x, 16x, or 32x. Empirical results (Section 3.2) show that this system is robust across different LLMs and caption styles, providing a general mechanism for content-aware adaptation.

[1]Anonymous Institution, Anonymous City, Anonymous Region, Anonymous Country. Correspondence to: Anonymous Author <anon.email@domain.com>.

Preliminary work. Under review by the International Conference on Machine Learning (ICML). Do not distribute.

Figure 1: **Content-Adaptive Tokenization.** CAT uses an LLM to determine the compression ratio from the image's text description and uses a nested VAE to generate latent features by dynamically routing the input.

To support variable-length representations, we design a nested variational autoencoder (VAE) architecture that routes intermediate encoder features to a shared latent block for generating Gaussian parameters of different shapes, enabling latent codes at multiple spatial scales. This design allows us to train a *single* model that supports multiple compression levels while maintaining architectural efficiency.

We train CAT on a diverse set of images using LLM-evaluated compression ratios and conduct extensive evaluations across nine datasets, covering natural scenes (ImageNet (Deng et al., 2009), COCO (Lin et al., 2015)), human faces (CelebA (Liu et al., 2015)), and detail-heavy domains such as text (ChartQA (Masry et al., 2022), GTSRB (Stallkamp et al., 2011), SVHN (Netzer et al., 2011)), textures, and satellite imagery. On large-scale natural images, CAT preserves high reconstruction quality while **reducing token usage by 18%** compared to fix-token baselines. On complex images, CAT achieves significantly better reconstruction quiality, **improving the rFID by 12% on CelebA, 17% on GTSRB, 20% on SVHN, and 39% on ChartQA** relative to fixed-token baselines. We also benchmark CAT on two critical downstream tasks:

- **Image classification:** CAT achieves the **highest linear probing accuracy** compared to all fixed-token baselines across five challenging datasets where prior work has shown that zero-shot models perform poorly (Ilharco et al., 2021). This highlights the quality of our content-adaptive latent representations. Besides, CAT consistently improves performance in full fine-tuning settings.

- **Text-to-Image Generation:** We integrate CAT into Latent Diffusion Transformers (DiTs) (Peebles and Xie, 2022). On class-conditional ImageNet generation, CAT **increases the training throughput by 23%**, achieving **better FIDs** than all fixed-ratio tokenizers trained under the same FLOPs. We note that CAT allows users to specify the desired token count at inference, enabling a flexible trade-off between computational cost and output quality, with more tokens typically yielding higher fidelity.

In summary, we propose CAT, an efficient and effective image tokenizer that enables content-adaptive compression

through an LLM-based evaluator and a nested VAE architecture. To the best of our knowledge, this is the first work to combine language-guided tokenization with adaptive representation, showing both performance and efficiency gain in image reconstruction, classification, and generation.

## 2. Related Work

**Image tokenization.** Existing tokenizers use diverse architectures and encoding schemes. Continuous tokenizers often utilize the VAE architecture (Kingma and Welling, 2014) to generate Gaussian distributions for sampling continuous latent features. Discrete tokenizers like VQ-VAE (van den Oord et al., 2018), RQ-VAE (Lee et al., 2022), MoVQ (Zheng et al., 2022), MAGVIT-v2 (Yu et al., 2024a), and FSQ (Mentzer et al., 2023) use quantization techniques to convert latent representations into tokens. VQ-GAN (Esser et al., 2020), ViT-VQGAN (Yu et al., 2021), and Efficient-VQGAN (Cao et al., 2023) further built on adversarial training to improve performance. Beyond methods that tokenize images into 2D grids, 1D tokenizers such as TiTok (Yu et al., 2024b) are proposed to enhance efficiency. While CAT is designed as a continuous 2D tokenizer, the proposed adaptive image encoding scheme can be applied to discrete and 1D tokenizers.

**Adaptive compression.** Traditional codecs like JPEG (Wallace, 1992) for images and H.264 (Wiegand et al., 2003) for videos apply varying levels of compression based on the input media, producing files of different sizes. In deep learning, patch dropout (Chen et al., 2023; Rao et al., 2021), patch merging (Yin et al., 2022; Bolya et al., 2023; Shen and Yang, 2021; Shen et al., 2024a), and sequence packing (Dehghani et al., 2023) are proposed for Vision Transformers (Dosovitskiy et al., 2020). Quadformer (Ronen et al., 2023) uses mixed-resolution patches to vary token count. However, these methods are tailored for visual understanding tasks and cannot be used for generation. A few recent works such as VAR (Tian et al., 2024) study multi-scale tokenization for generation. Nonetheless, these works are not adaptive to image content.

| Tol=0.0001 | Tol=0.0005 | Tol=0.0015 |
| --- | --- | --- |

| Metric | Pearson $r$ |
| --- | --- |
| JPEG | 0.31 |
| MSE | 0.36 |
| LPIPS | 0.23 |
| Caption | **0.55** |

Figure 2: **L:** Max acceptable compression ratios for different $\tau$. **R:** Correlation with max acceptable compression for $\tau = 0.0015$.

Adaptive tokenizers for image generation remain relatively underexplored. ElasticTok (Yan et al., 2024) employs random masking to drop the tail tokens during training. ALIT (Duggal et al., 2024) iteratively distills 2D tokens into 1D to reduce the token count. DQ-VAE (Huang et al., 2023) leverages information-density for dynamic representation. However, all of these methods (1) require an input image to determine the token count, limiting their use in generation settings where only text is available; and (2) overlook image complexity during training. In contrast, we enable adaptive compression directly from textual descriptions without observing the image. We also explicitly train the tokenizer with complexity-aware supervision. A concurrent work, TexTok (Zha et al., 2024), explores language-guided tokenization by supplying the caption embeddings to the VAE. However, it is not designed for adaptive representation.

**Multi-scale network design.** Our work is also related to designing neural networks for multi-scale feature extraction. Inspired by U-Net (Ronneberger et al., 2015) and Matryoshka networks (Kusupati et al., 2022; Cai et al., 2024; Gu et al., 2024), we incorporate skip connections into the VAE to support multi-ratio compression in a single forward pass. Parallel work explores transformer-based multi-scale architectures (Nash et al., 2022; Shen et al., 2023; 2024b; Roberts et al., 2021; Yuan et al., 2021b; Shen et al., 2024c; Hu et al., 2024). To the best of our knowledge, our nested design offers the simplest yet effective solution for generating multi-scale latents via VAEs without additional architectural or computational overhead, while achieving strong results (Section 4).

## 3. Method

In this section, we present CAT for adaptive image tokenization. We begin by motivating our caption-based evaluator for image complexity estimation. Then, we describe the nested VAE architecture.

### 3.1. Proof of Concept

**How much can we actually compress?** A key question in this work is to determine how much an image can be compressed without significant loss of quality. To explore this, we study the reconstruction performance of existing tokenizers under various compression ratios. We take the open-source LDM tokenizers [1] with 8x, 16x and 32x compression ratios and compute their reconstruction mean squared error (MSE) on 41K COCO 2014 (Lin et al., 2015) images with resolution 512. We find that for 28.3% of the images, 32x compression results in less than a 0.001 MSE increase compared to 8x, while reducing the token count by a factor of 16. We also compute the best MSE among all compression ratios for each image and determine the maximum acceptable compression ratio under a tolerance $\tau$ ($\arg\max_{ratio} \text{MSE}_{ratio} - \text{MSE}_{best} < \tau$). Fig. 2 (left) shows that 56% of the images can be compressed at least to 16x with negligible (0.0001) increase in MSE[2]. That is, a large portion of natural images can be compressed more aggressively to save compute.

On the other hand, our visual inspection reveals that images with fine-grained elements like text have significantly worse reconstruction quality at 32x compression compared to 8x compression (e.g., see row 3 and 4 in Figure 3). This suggests that more tokens are required to accurately reconstruct low-level details. The above results provide strong motivation for developing an adaptive tokenizer.

**Limitations of existing complexity metrics.** Next, we want to identify a metric for predicting an image's optimal compression ratio. We explore two existing options: (1) metrics produced by traditional codecs, such as the JPEG file size; (2) metrics based on pretrained VAEs, such as the reconstruction MSE and LPIPS distance (Zhang et al., 2018). We use Stable Diffusion's sd-vae-ft-mse (AI, 2022) for this analysis. We compute these metrics on COCO and analyze their correlation with the maximum acceptable compression ratio under 0.0015 tolerance. However, Table 2 shows that the Pearson $r$'s are relatively low. Statistically, these metrics are not highly correlated with an image's complexity.

We also manually inspect images with large JPEG sizes and MSEs. We note that images featuring repetitive patterns, such as grass, forests, and animals like giraffes and zebras consistently show high complexity metrics. Indeed, JPEG compression can be inefficient for images with sharp edges and high contrast. A single-pixel shift in a zebra image can toggle pixel values between black and white, significantly increasing the reconstruction error. However, as Figure 3 (left) show, large JPEG sizes or MSEs do not always notably affect visual quality. For example, we can easily recognize the zebra and may not perceive the differences resulting from various compression ratios.

---

[1]LDM (Rombach et al., 2021) released a series of VAE tokenizers with diverse compression ratios trained in a controlled setting. Most other tokenizers like `stabilityai/sd-vae-ft-mse` only have one compressed ratio.

[2]The average MSE across all images for 8x LDM VAE is 0.0039, so a 0.0001 tolerance should be acceptable.

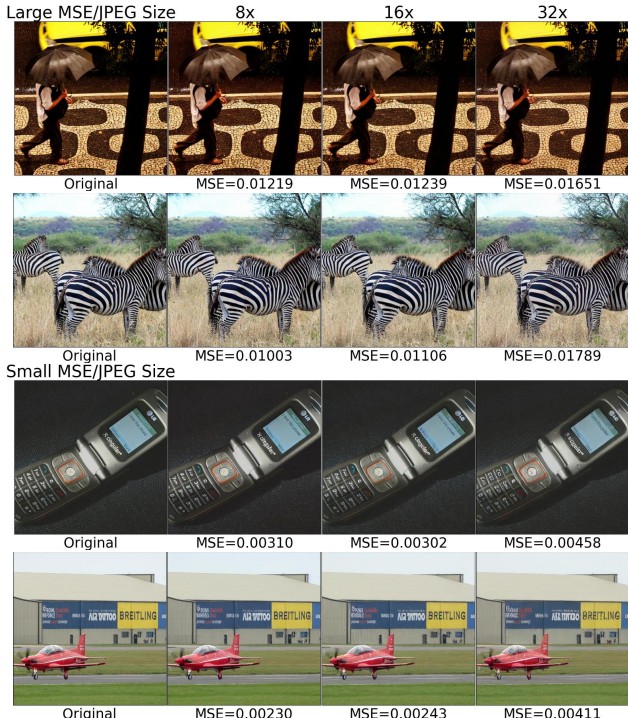

Figure 3: **Existing metrics can misjudge image complexity.** Text-heavy images that are difficult to difficulty to reconstruct (note the distortion in the bottom two rows) are considered as easy by existing metrics.

Conversely, images with small JPEG sizes and low MSEs can have poor fidelity. For example, as Figure 3 (right) shows, distortions in images containing elements sensitive to human perception, such as text, numbers, and human faces, can drastically reduce visual quality. Despite this, these images have low reconstruction errors since the critical elements occupy only small portions of the images.

Thus, existing metrics fail to capture details crucial to human perception. In contrast to the predicted complexity, we actually want to use a large compression ratio for zebra images, and a small ratio for the phone images. Beyond this, all considered metrics require images as input and cannot be used for text-to-image generation tasks, where no image is available. Given all these limitations, we seek a new complexity metric that is independent of pixel data and aligns with human perception. We note that it is impractical to use the tolerance-based compression ratio in Figure 2 as the complexity metric because it requires at least three model calls to get the MSEs, making it computationally expensive.

### 3.2. Complexity Evaluation via Captions and LLMs

Image generation typically involves users providing a prompt that describes the desired image content. Inspired by this use case, we propose to use the text description of an image to evaluate its complexity.

We propose a 3-stage evaluation pipeline: (1) obtaining the text description, (2) prompting a LLM for a complexity score, and (3) classifying the score into a compression ratio. The text description includes both the image caption and answers to a set of pre-defined queries of the form "*Are there [obj]?*" For example, $obj \in \{human\ faces, text\}$ can be used to align with human perceptions. During training, when image data are available, we use a vision language model (VLM) to generate the captions and answers. During inference, users need to provide the necessary textual information.

We prompt a language model with the text description to generate an integer score ranging from 1 to 10, where higher scores indicate greater complexity. To ensure scoring consistency, we design a detailed rubric that instructs the model to consider factors including semantic complexity (objects, scenes), visual complexity (color, texture), and perceptual complexity. We also provide in-context examples for each score. The complete prompt is provided in Appendix B.

Based on the complexity score, each image is classified into one of three compression ratios: 8x, 16x, or 32x, with higher complexity scores corresponding to lower ratios. We choose these options as they are widely used in existing tokenizers (Rombach et al., 2021) and provide meaningful variation in token counts. Then, we implement a thresholding scheme to divide the scores into three intervals: $[1, a]$, $(a, b]$, and $(b, 10]$, where $a < b \in \mathbb{Z}^+$ are selected to achieve an average compression ratio of 16x across all training data to enable fair comparison with fixed 16x tokenizers. Formally, denote the training distribution as $\mathcal{X}$, input resolution as $r$, the compression ratio of an image $x \in \mathcal{X}$ as $f(x) \in \{f_1 = 8, f_2 = 16, f_3 = 32\}$, the target average compression ratio as $\bar{f} := 16$. We set $a, b$ by:

$$\mathbb{E}_{x \in \mathcal{X}}\left[\frac{r^2}{f(x)^2}\right] \approx \sum_{x \in \mathcal{D}} p(f(x)) \frac{r^2}{f(x)^2} \approx \frac{r^2}{\bar{f}^2} \qquad (1)$$

While multiple threshold configurations can achieve the target average compression ratio, our experiments in Section 4.2 demonstrate that a diverse distribution of ratios yields better performance. For the specific training dataset and thresholds used in our experiments, see Section 4.

**Robustness testing and bias mitigation.** To evaluate the robustness of our caption-based complexity evaluator, particularly under different model choices and caption styles, we conduct ablation studies on diverse datasets such as ImageNet (Deng et al., 2009), CelebA (Liu et al., 2015) and EuroSAT (Helber et al., 2019). We compare two pipelines: (1) a unified pipeline using LLaVA1.5 7B (Liu et al., 2023) for both captioning and scoring in a single pass, and (2) a separated pipeline using InstructBLIP (Dai et al., 2023a) for captioning and either Qwen2.5 7B (Yang et al., 2024) or LLaMA3.1 8B Instruct (Dubey et al., 2024) for scoring. Additionally, within the unified pipeline, we test variants that explicitly prompt the model to produce longer versus

shorter captions. As shown in Appendix Table 10, all configurations result in similar compression distributions. We attribute this robustness to (1) our carefully designed scoring rubrics and in-context examples, and (2) the LLM's ability to infer content complexity is more influenced by the semantic content of the caption than its wording. These findings suggest that our evaluator is robust to model choices and captioning variations. Given its simplicity and efficiency, we adopt the unified pipeline in all subsequent experiments.

**High correlation with visual complexity.** We verify that our caption score provides reliable estimates of the optimal compression ratio. Similar to Section 3.1, we compute the correlation between complexity scores and maximum acceptable compression ratios on COCO. Our metric achieves the highest Pearson $r$ among all options (Table 2) and 62.39% exact agreement on compression ratio selection. Manual inspection also confirms that perceptually challenging images receive high complexity scores. We note that during development, we also tested other caption-derived metrics such as caption length, but it does not perform as well as LLM-based scoring (Appendix Table 9).

**Minimal captioning overhead.** When text information is available, our pipeline requires only a single LLM call to obtain the complexity score. During training, where only images are provided, we generate the caption, query responses, and complexity score within a single inference pass, keeping the evaluation overhead minimal. Given the efficiency of modern inference engines like vLLM (Kwon et al., 2023), the cost of complexity evaluation is negligible compared to the overall compute required for training the tokenizer and downstream models.

### 3.3. Nested VAE for Adaptive Compression

To reduce the tokenizer's training and storage costs, we introduce a nested structure to the standard VAE architecture (Kingma and Welling, 2014) to enable multiple compression ratios within a single model. In standard VAE architecture, the encoder consists of multiple downsampling blocks followed by an attention-based middle block. The decoder consists of an attention-based middle block followed by upsampling blocks. This symmetrical design resembles U-Net (Ronneberger et al., 2015) and Matryoshka networks (Kusupati et al., 2022) for multi-scale feature extraction. Inspired by these works, we leverage the intermediate outputs of the downsampling blocks to enable adaptive compression (Figure 1). We describe the proposed architecture below.

**Skip connection with channel matching.** Denote the feature shape under the largest compression ratio as $(c_3, \frac{r}{f_3}, \frac{r}{f_3})$, where $c_3$ is the channel dimension. We observe that, in the standard VAE encoder, the spatial dimension of the intermediate outputs from the downsampling blocks decreases by a factor of 2 with each additional block. This

means that the output of the second-to-last downsampling block has shape $(c_2, \frac{r}{f_2}, \frac{r}{f_2})$, and the output of the third-to-last downsampling block has shape $(c_1, \frac{r}{f_1}, \frac{r}{f_1})$. Then, a natural thought is to directly route these intermediate outputs to the middle block to generate latent features. However, since the channel dimensions of these intermediate outputs vary, we incorporate ResNet blocks (He et al., 2015) for channel matching. Let the latent channel dimension of the VAE be $c$. Applying channel matching enables us to transform intermediate features of shape $(c_n, \frac{r}{f_n}, \frac{r}{f_n})$ to $(c, \frac{r}{f_n}, \frac{r}{f_n})$ for $n = 1, 2, 3$. This will be the shape of the latent parameters.

For decoder, we similarly add skip connection with channel matching and use the decoder middle block's output as the input to the corresponding upsampling block, i.e., for compression ratio $f_n$, we bypass the first $n - 1$ upsampling blocks so the final output has the same size as the original image.

**Shared mean/variance parametrization.** Features after channel matching are directed to the middle block to generate the latent parameters. In CAT, we share the middle block for all compression ratios to maintain scale consistency of the parameterized mean and variance. The convolutional design of the middle block allows it to process inputs of varying spatial dimensions, as long as the channel dimension is aligned. Thus, for all $n \in \{1, 2, 3\}$, the mean $\mu_n$, variance $\sigma_n^2$, and samples $z_n$ of the Gaussian distribution all have shape $(c, \frac{r}{f_n}, \frac{r}{f_n})$, i.e., the input compressed at $f_n$.

**Increasing parameter allocation for shared modules.** Images assigned larger compression ratios do not go through the later downsampling blocks and are directed straight to the middle block. The middle block is thus tasked with handling multi-scale features. To improve its capacity, we allocate more parameters to the middle block by increasing the number of attention layers.

### 3.4. Training

While existing adaptive tokenizers like ElasticTok (Yan et al., 2024) do not consider the varying complexity of training data, we explicitly incorporate content complexity into training to learn feature extraction at different granularity. For each training example, we first obtain the compression ratio from the LLM. Then, the image is processed only by the layers dedicated to its compression ratio.

Similar to prior works (Kingma and Welling, 2014; Esser et al., 2020), we use a joint objective that minimizes reconstruction error, Kullback-Leibler (KL) divergence, and perceptual loss. Specifically, we use $\mathcal{L}_1$ loss for pixel-wise reconstruction. To encourage the encoder output $z$ towards a normal distribution, KL-regularization is added: $\mathcal{L}_{\text{KL}}(z) := \mathbb{KL}(q_\theta(z|x)\|p(z))$, where $\theta$ is the encoder parameters and $p(z) \sim \mathcal{N}(0, \mathbf{I})$. The perceptual loss consists

Table 1: **Test data distribution.** CAT applies larger compression to natural images and smaller ratios to CelebA and ChartQA.

| Datasets | Method | 8x | 16x | 32x | Avg Latent Dim | Avg Rate |
|---|---|---|---|---|---|---|
| COCO | **CAT (Ours)** | 9% | 54% | 37% | 31.87 | 16.07x |
| | JPEG | 10% | 54% | 36% | 32.43 | 15.79x |
| ImageNet | **CAT (Ours)** | 6% | 49% | 45% | 29.32 | 17.46x |
| | JPEG | 9% | 49% | 42% | 31.24 | 16.39x |
| CelebA | **CAT (Ours)** | 17% | 83% | 0% | 39.29 | 13.03x |
| | JPEG | 0% | 0% | 100% | 16 | 32x |
| ChartQA | **CAT (Ours)** | 96% | 4% | 0% | 63.02 | 8.12x |
| | JPEG | 0% | 3% | 97% | 16.61 | 30.82x |

of the LPIPS similarity (Zhang et al., 2018) and a loss based on the internal features of the MoCo v2 model (He et al., 2020). Beyond these, we train our tokenizer in an adversarial manner (Goodfellow et al., 2014) using a patch-based discriminator $\psi$, which adds a GAN loss $\mathcal{L}_{\text{GAN}}(x, \hat{x}, \psi)$. Thus, our overall objective is:

$$\mathcal{L} = \min_{\theta} \max_{\psi} \; \mathbb{E}_{x \in \mathcal{X}} \Big[ \mathcal{L}_{\text{rec}}(x, \hat{x}) + \beta \mathcal{L}_{\text{KL}}(z)$$
$$+ \gamma \mathcal{L}_{\text{perc}}(\hat{x}) + \delta \mathcal{L}_{\text{GAN}}(x, \hat{x}, \psi) \Big], \quad (2)$$

where $\beta, \gamma, \delta$ are loss weights. We provide our training code in the supplementary material.

# 4. Image Reconstruction

We first evaluate CAT's reconstruction quality and analyze various design choices via ablation studies.

**Training details.** We use a nested VAE with six downsampling blocks and 187M parameters. For our main results (Table 2), we set the latent channel $c$ to 16. We study its effect as an ablation study in Section 4.2. For training data, we use 380M licensed Shutterstock images with 512x512 resolution. After obtaining the complexity scores, we find that two sets of thresholds, $(a, b) \in \{(4, 7), (2, 8)\}$, both achieve an average compression ratio of 16x. We select $(4, 7)$ for our main experiments because it leads to a more diverse distribution and better empirical results (see ablation studies in Section 4.2). All models including the baselines are trained using a global batch size of 512 on 64 NVIDIA A100 GPUs for 1M steps. Other architecture and training details can be found in Appendix E.

**Evaluation datasets.** We use four representative datasets: COCO (Lin et al., 2015) and ImageNet (Deng et al., 2009) for natural images, CelebA (Liu et al., 2015) and ChartQA (Masry et al., 2022) for perceptually challenging images. Table 1 shows the compression ratio distributions.

**Baselines.** We compare against fixed-token baselines that use the same VAE architecture but without the nested structure. We further study the effect of caption complexity by training a nested VAE using JPEG size as the complexity metric. All baselines have average 16x compression. To

our knowledge, none of existing adaptive tokenizers (e.g., ElasticTok, ALIT) report quantitative results on the datasets we use, so we do not compare with them. ALIT only shows an rFID of 8.03 on ImageNet100. See Appendix E.3 for more baseline details.

## 4.1. Main Results

**Better reconstruction for complex images, higher efficiency for natural images.** Table 2 presents the reconstruction FID (rFID), LPIPS, and PSNR (Horé and Ziou, 2010) on four datasets. Comparing CAT with the fixed 16x baseline, our method significantly outperforms the baselines across all metrics on CelebA and ChartQA, **improving the rFID by 12% on CelebA and 39% on ChartQA**. Our **ChartQA rFID even surpasses the fixed 8x baseline**, likely because the LLM evaluator can effectively identify rich visual details in these datasets and assign lower compression ratios accordingly (Table 1). On COCO and ImageNet, CAT generally outperforms the baselines, with only a slight drop in rFID on ImageNet. However, On ImageNet, CAT achieves an average compression ratio of 17.46x, which means we use **18% fewer tokens** to represent the dataset compared to the 16x baseline,

Figure 4 shows qualitative examples of progressive reconstruction quality as we manually increase the token count and reduce the compression ratio to represent each image. We highlight the compression ratio predicted by CAT in red. Different visual inputs need different ideal compression ratios. Natural images with fewer objects and simpler patterns can be accurately reconstructed at 32x, whereas complex images with visual details require lower compression. Thus, the caption-based CAT reconstruction has comparable quality to the fixed 16x baseline on natural images but surpasses it on text-heavy images. We include more visualization in Appendix E.4. We also report CAT's performance with uniform 16x compression in Appendix E.5.

Lastly, while most prior works do not evaluate on diverse datasets like ChartQA, we include a side-by-side comparison with popular fixed-ratio tokenizers on ImageNet in Table 3. This table is for reference due to differing training setups (we refer the readers to Table 2 for fully comparable baselines). Still, the results highlight that CAT achieves competitive performance by adapting compression ratios to image content.

Table 3: ImageNet-512 reconstruction.

| ImageNet | rFID |
|---|---|
| MaskGIT-VQGAN (Chang et al., 2022) | 1.97 |
| TiTok-B-128 (Yu et al., 2024b) | 1.52 |
| LFQ (Yu et al., 2024a) | 1.22 |
| TexTok (Zha et al., 2024) | 0.73 |
| LDM (Rombach et al., 2021) | 0.53 |
| CAT | 0.46 |

**Caption complexity outperforms JPEG metric.** We further compare CAT against training the same adaptive architecture but using JPEG size as the complexity metric. Table 2 shows that **CAT achieves better rFID, LPIPS, and PSNR across all datasets**. While both tokenizers have

Table 2: **Reconstruction results.** All models have latent channel $c = 16$.

| Average Compression | | | COCO | | | ImageNet | | | CelebA | | | ChartQA | | |
|---|---|---|---|---|---|---|---|---|---|---|---|---|---|---|
| | | | rFID↓ | LPIPS↓ | PSNR↑ | rFID↓ | LPIPS↓ | PSNR↑ | rFID↓ | LPIPS↓ | PSNR↑ | rFID↓ | LPIPS↓ | PSNR↑ |
| 8 | Fixed | 8x | 0.48 | 0.10 | 30.95 | 0.24 | 0.095 | 33.86 | 1.86 | 0.028 | 45.36 | 8.21 | 0.019 | 36.98 |
| 16 | Fixed | 16x | 0.66 | 0.16 | 29.79 | **0.38** | **0.15** | 30.45 | 2.25 | 0.059 | 41.84 | 8.67 | 0.029 | 33.48 |
| | Adaptive | JPEG | 0.72 | 0.17 | 30.11 | 0.51 | 0.16 | 30.61 | 6.57 | 0.14 | 36.47 | 10.17 | 0.048 | 31.54 |
| | Adaptive | **CAT (Ours)** | **0.65** | **0.15** | 30.19 | 0.46 | **0.15** | **30.62** | **1.97** | **0.051** | **42.43** | **5.27** | **0.021** | **36.45** |
| 32 | Fixed | 32x | 1.18 | 0.26 | 26.93 | 0.81 | 0.25 | 27.48 | 6.10 | 0.16 | 36.35 | 10.79 | 0.045 | 30.99 |

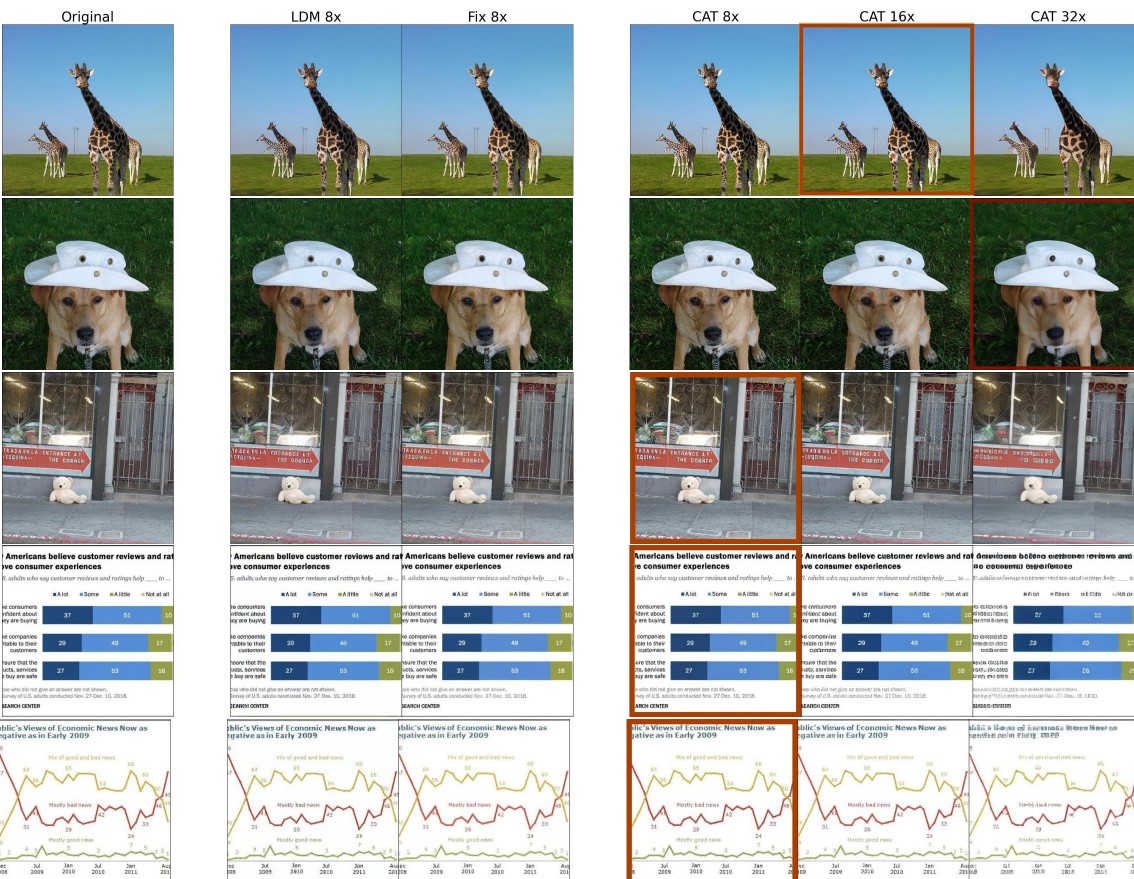

Figure 4: **Qualitative reconstruction examples.** We highlight the compression ratio selected by our caption complexity in red. On simpler images (top two rows), adjusting the compression ratio does not significantly affect quality. On more complex images (bottom three rows), the impact is substantial.

similar training compression ratio distribution, the test-time compression ratio distribution varies significantly (Table 1). Notably, since JPEG size often cannot capture perceptually important factors (see Section 3.1), nearly all images in CelebA and ChartQA are assigned 32x compression. Thus, CAT significantly outperforms JPEG on these two datasets, showing the effectiveness of caption-based metric and LLM evaluation in determining image intrinsic complexity.

### 4.2. Ablation Studies

**Benefits of diverse compression ratios.** We explore several design choices for our tokenizer. First, we study how

the distribution of compression ratios affects overall reconstruction. To achieve an average compression ratio of 16, we could set the thresholds $(a, b)$ to either $(4, 7)$ or $(2, 8)$. $(4, 7)$ yields a more diverse distribution of compression ratios, whereas $(2, 8)$ results in a concentrated distribution similar to a fixed 16x tokenizer, making it less interesting. Table 4 shows that $(4, 7)$ produces better reconstruction metrics across all datasets due to the diversity in compression ratios ensures that all parts of the model are fully trained. Hence, we adopt $(4, 7)$ as the thresholds for CAT.

**Effect of latent channel dimension.** We also vary the latent channel $c$ to study its effect. Table 5 shows that a larger $c$ improves reconstruction results. However, similar

Table 4: **Ablation on thresholds.** The (4, 7)-setting with more diverse training distribution achieves generally better performance.

| $(a, b)$ | COCO | ImageNet | CelebA | ChartQA |
|---|---|---|---|---|
| (4, 7) | **0.65** | 0.46 | **1.97** | **5.27** |
| (2, 8) | 0.67 | **0.43** | 2.58 | 7.70 |

Table 5: **Ablation on latent channel.** Increasing latent channel $c$ improves rFID across all four evaluated datasets.

| $c$ | COCO | ImageNet | CelebA | ChartQA |
|---|---|---|---|---|
| 4 | 1.66 | 1.10 | 5.83 | 9.13 |
| 8 | 1.03 | 0.60 | 4.54 | 7.95 |
| 16 | **0.65** | **0.46** | **1.97** | **5.27** |

Table 6: **More reconstruction and classification results.** CAT achieves better reconstruction and classification results.

| | DTD | EuroSAT | GTSRB | SUN397 | SVHN |
|---|---|---|---|---|---|
| *Reconstruction FID ($\downarrow$)* | | | | | |
| LDM VAE 16x | 7.86 | 7.04 | 1.22 | 1.95 | 1.76 |
| Fix 16x | 7.29 | 6.26 | 1.38 | 1.95 | 2.11 |
| **CAT (Ours)** | **7.23** | **5.45** | **1.14** | **1.93** | **1.69** |
| *Linear Probing Top-1 Accuracy (%, $\uparrow$)* | | | | | |
| LDM VAE 16x | 53.81 | 75.38 | 70.29 | 63.70 | 64.87 |
| Fix 16x | 50.42 | 78.08 | 70.07 | 62.92 | 65.07 |
| **CAT (Ours)** | **54.51** | **78.21** | **71.28** | **64.16** | **66.39** |
| *Fine-Tuning Top-1 Accuracy (%, $\uparrow$)* | | | | | |
| LDM VAE 16x | **75.91** | 97.00 | 90.61 | **79.89** | 83.05 |
| Fix 16x | 71.96 | 92.46 | 95.07 | 78.77 | 85.93 |
| **CAT (Ours)** | 74.11 | **98.00** | **95.32** | 78.45 | **86.13** |

to previous work (Rombach et al., 2021; Dai et al., 2023b), we observe a reconstruction-generation trade-off: while increasing $c$ is beneficial for reconstruction, it does not necessarily improve second-stage generative results. We elaborate more on this in Section 6.

## 5. Classification on Diverse Datasets

While CAT demonstrates strong reconstruction performance on COCO, ImageNet, CelebA, and ChartQA, these datasets represent only a subset of image domains. To thoroughly assess CAT's capabilities in diverse domains and tasks beyond reconstruction, we follow OpenCLIP (Ilharco et al., 2021) and test CAT's classification performance on five datasets that feature details beyond text and human faces: DTD (Cimpoi et al., 2014) (textures), EuroSAT (Helber et al., 2019) (satellite images), GTSRB (Stallkamp et al., 2011) (traffic signs), SUN397 (Xiao et al., 2016) (indoor and outdoor scenes), and SVHN (Netzer et al., 2011) (street numbers). We examine two settings: (1) linear probing, where we freeze the encoder and train only a linear layer on top of the latent features; and (2) fine-tuning both the encoder and classification head. See Appendix F for experiment details.

As shown in Table 6, CAT consistently outperforms fixed-ratio baselines across both settings, achieving the **best top-1 accuracy on all five datasets in the linear probing setup**. This highlights the quality and generalizability of CAT's latent representations, which transfer effectively to downstream classification tasks even without fine-tuning. In addition, CAT maintains strong reconstruction performance on these datasets, **improving the rFID by 13% on EuroSAT, 17% on GTSRB, and 20% on SVHN** (Table 6, top rows). These results further confirm that our caption-based complexity metric supports learning representations that balance both compression quality and downstream utility.

## 6. Image Generation

Lastly, we evaluate CAT on text-to-image generation to show that (1) its adaptive design does not compromise generation quality; (2) by using fewer tokens to represent the training data, CAT actually enables more efficient learning, leading to stronger generative models under the same compute budget.

**Setup.** Given CAT's ability to produce variable-length token sequences, we integrate it with Diffusion Transformer (DiT) (Peebles and Xie, 2022), which handles adaptive token representations naturally. DiT takes the noised latent features as input, applies patching for downsampling, and uses a transformer architecture to predict the added noise. Following (Peebles and Xie, 2022), we work with class-conditional generation on ImageNet-512, leveraging DiT-XL with a patch size of 2.

During training, each image is processed through the CAT pipeline to determine its compression ratio, resulting in variable-length latent representations tailored to its complexity. That is, *each example can have different number of latent tokens.* For inference, we obtain a class-level compression ratio by providing a textual description of the form "*this is an image of [label]*" to the LLM evaluator, which predicts the appropriate compression ratio for that class. For example, if the evaluator suggests a 16x compression, we generate $(\frac{512}{16 \cdot 2})^2 = 256$ tokens, where the 2 in the denominator accounts for DiT's patching. The generated tokens are then decoded by CAT to reconstruct 512x512 images. Following (Peebles and Xie, 2022), we report FID (Heusel et al., 2018), Sliding FID (Ding et al., 2023), Inception Score (Salimans et al., 2016), precision and recall (Kynkäänniemi et al., 2019) on 50K images generated with 250 DDPM steps and classifier-free guidance (Ho and Salimans, 2022). We note that we use the same compression ratio per class predicted by LLMs mainly for benchmarking purpose. In reality, CAT allows users to flexibly specify the desired token count at inference time, as we will show later.

Table 7: **ImageNet-512 generation results**. All tokenizers have average 16x compression ratio. "rFLOPs" means relative FLOPs.

| | | FID↓ | sFID↓ | IS↑ | Precision↑ | Recall↑ | Eval rFLOPs↓ |
|---|---|---|---|---|---|---|---|
| Fixed | DiT + LDM VAE | 10.03 | 16.88 | 114.84 | 0.65 | **0.50** | 1× |
| | DiT + Fixed 16x | 4.78 | 11.81 | 187.47 | 0.72 | 0.49 | 1× |
| Adaptive | DQ-Transformer (Huang et al., 2023) | 5.11 | - | 178.2 | - | - | - |
| | **DiT + CAT (Ours)** | **4.56** | **10.55** | **191.09** | **0.75** | 0.49 | **0.82×** |

Table 8: Larger channel $c$ is not always better for generation.

| $c$ | FID↓ | sFID↓ | IS↑ | Precision↑ | Recall↑ |
|---|---|---|---|---|---|
| 4 | 5.12 | 11.12 | 152.39 | 0.72 | 0.48 |
| 8 | **4.38** | **10.31** | 181.03 | **0.76** | 0.48 |
| 16 | 4.56 | 10.55 | **191.09** | 0.75 | **0.49** |

**Baselines.** We compare against DiT-XL paired with the open-source 16x LDM VAE and the fixed 16x tokenizer we trained ourselves. All models are trained on 16 NVIDIA H100 GPUs for 400K steps, using a global token batch size of 262,144, which is equivalent to 1024 images at 16x compression. See Appendix G for more implementation and baseline details.

As shown in Table 7, DiT-CAT achieves the **best FID, sFID, IS, and precision** among all DiT baselines trained with the same computational resources. We attribute this performance to two factors. First, using fewer tokens for simpler images improves processing efficiency, allowing for more extensive training within the same computational budget. In fact, the average token count per *training* image for DiT-CAT is 197.44. Compared to the 256 tokens used by fixed 16x tokenizers, CAT achieves a **23% reduction in token count**, allowing the model to process more images within the same training budget. Second, adaptively allocating representation capacity also enables more effective modeling of complex images, as richer visual details are better preserved through the use of additional tokens.

As discussed earlier, we use our LLM evaluator to obtain the generation token count using pre-defined captions mainly because we want to automate the evaluation process. In reality, users can flexibly set the token count based on their computational budget. To explore this, we manually vary the token count during generation with DiT-CAT and observe that FID-50K improves from 5.83 (64 tokens) to 5.02 (256 tokens) and 4.12 (1024 tokens), confirming that more tokens lead to higher-quality images. Qualitative examples in Figure 5 further support this observation. Thus, CAT enables a controllable trade-off between efficiency and generation quality. For more visualization, see Appendix G.4.

Lastly, recall that we trained tokenizers with different latent channels in Section 4.1. Table 8 shows the generation performance. While larger $c$ is better for reconstruction, it does not benefit generation. CAT with $c = 8$ achieves the best

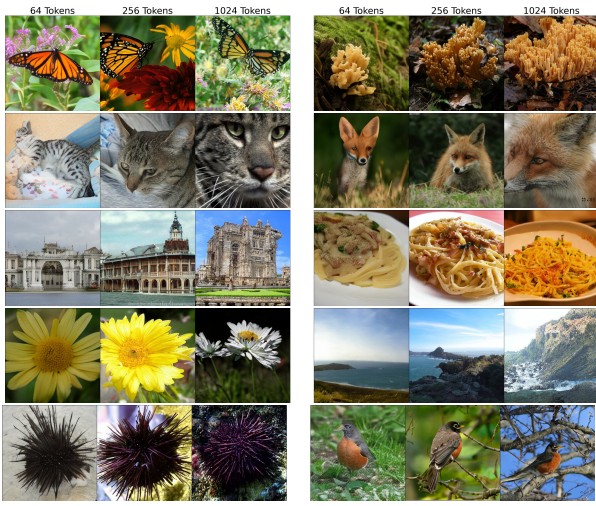

Figure 5: Increasing token count (left→right) for CAT leads to better image quality and higher complexity.

FID across all experiments. This observation agrees with prior work (Rombach et al., 2021) and highlights the importance of choosing an appropriate $c$. We leave exploring $c$'s impact on downstream task for future work.

# 7. Conclusion

We propose an adaptive image tokenizer, CAT, that allocates different number of tokens to images based on the content complexity derived from text description. Our experiments show that CAT improves the quality and efficiency of image representation on a variety of downstream tasks.

**Limitations and Future Work.** Nested VAE is a natural extension of the VAE architecture but is constrained to predefined compression ratios that scale by factors of 2. An intriguing future direction would be to enable more flexible compression ratios by transitioning to transformer backbones and dynamically adjusting token counts. Besides, images contain diverse global and local information. While CAT addresses global complexity by increasing token allocation for images with intricate details, further efficiency improvements could come through local tokenization—allocating more tokens specifically to detailed regions while reducing tokens for simpler areas. Besides, an ideal pipeline would enable LLMs to automatically identify perception-critical elements without relying on pre-defined queries. This would be an important next step.

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

## A. Broader Impact

This work introduces CAT, a content-adaptive tokenizer designed to improve the efficiency and flexibility of image modeling. By enabling dynamic compression based on image content, CAT has the potential to reduce computational and energy costs in large-scale generative and classification systems, contributing to more sustainable AI practices. However, as with any generative model, misuse of this technology for creating misleading or harmful content remains a concern. We encourage responsible deployment, including proper safeguards and usage guidelines, particularly in sensitive domains such as media generation, surveillance, or automated decision-making.

## B. Prompt for LLM Scorer

Our caption complexity pipeline works as follows:

Step 1: Use a VLM to generate image description, with the following prompts:

- What's in the image? → Caption

- Are there text or numbers in the image? → Yes/No.

- Are there faces in the image? → Yes/No.

Step 2: Use the same VLM or a separate LLM to generate the complexity score with the prompt:

Given the description of a 512px image, determine its complexity based on the following factors:
1. Number of distinct objects
2. Color variance
3. Texture complexity
4. Foreground and background
5. Symmetry and repetition
6. Human perception factors, like the presence of human faces or text
You will be given the caption, whether there are text or numbers, and whether there are faces in the image. Assign a complexity score such that a higher number means the image is more complex. Note that text and facial details are intrinsically complex because they are crucial to human perception. Here are some examples for scoring:
- Score 1: A plane in a sky
- Score 2: A t-shirt with a emoji on it
- Score 3: A dog lying on the grass
- Score 4: A woman skiing in the snow
- Score 5: Two kids walking on the beach
- Score 6: A dinning table full of food
- Score 7: A close-up shot of a old man
- Score 8: Many people gathering in the stadium
- Score 9: Newspapers or graphs with text and numbers
Now determine the complexity for the caption:
[*Insert caption here*]
[*Insert one of the following based on the Yes/No questions:*
*- There are text visible in the image. There are also facial details.*
*- There are text visible in the image, but there is no human face.*
*- There is no obvious text in the image, but there are facial details.*
*- There is no text or human face in the image.* ]
Respond with "Score: ? out of 9", where "?" is a number between 1 and 9. Then provide explanations.

Note these two steps can be combined into a single inference call.

# C. Other Complexity Metrics

Table 9: We also tried various other metrics. However, they are less effective (e.g., caption length) compared to our LLM-based score.

| Metric | Pearson $r$ |
|---|---|
| JPEG | 0.31 |
| MSE | 0.36 |
| LPIPS | 0.23 |
| Caption Length | 0.33 |
| CAT (Ours) | **0.55** |

## D. Compression Ratio Distributions with Different LLMs

Table 10: To study whether our caption score is robust to LLM choice, we test multiple captioners and LLMs for scoring. The fact that these combinations generate similar score and compression ratio distributions show that our scoring method is robust.

|  | Scoring Model | 8x | 16x | 32x | Avg Rate |
|---|---|---|---|---|---|
| ImageNet | InstructBLIP + Llama3.1-8B-Instruct | 6% | 49% | 45% | 17.43 |
|  | InstructBLIP + Qwen2.5-7B-Instruct | 2% | 70% | 28% | 17.35 |
|  | LLaVA1.5 7B | 2% | 65% | 33% | 17.75 |
|  | LLaVA1.5 7B (Longer Caption) | 3% | 62% | 35% | 17.58 |
| CelebA | InstructBLIP + Llama3.1-8B-Instruct | 17% | 83% | 0% | 13.02 |
|  | InstructBLIP + Qwen2.5-7B-Instruct | 15% | 70% | 15% | 13.83 |
|  | LLaVA1.5 7B | 5% | 95% | 0% | 14.92 |
|  | LLaVA1.5 7B (Longer Caption) | 6% | 94% | 0% | 14.72 |
| DTD | InstructBLIP + Llama3.1-8B-Instruct | 0% | 40% | 60% | 21.57 |
|  | InstructBLIP + Qwen2.5-7B-Instruct | 0% | 38% | 62% | 21.87 |
|  | LLaVA1.5 7B | 0% | 41% | 59% | 21.42 |
|  | LLaVA1.5 7B (Longer Caption) | 2% | 44% | 54% | 19.76 |
| EuroSAT | InstructBLIP + Llama3.1-8B-Instruct | 3% | 30% | 67% | 20.87 |
|  | InstructBLIP + Qwen2.5-7B-Instruct | 3% | 25% | 72% | 21.57 |
|  | LLaVA1.5 7B | 2% | 22% | 76% | 22.85 |
|  | LLaVA1.5 7B (Longer Caption) | 4% | 23% | 72% | 21.19 |
| SUN397 | InstructBLIP + Llama3.1-8B-Instruct | 6% | 73% | 21% | 15.82 |
|  | InstructBLIP + Qwen2.5-7B-Instruct | 2% | 78% | 20% | 16.77 |
|  | LLaVA1.5 7B | 3% | 80% | 17% | 16.30 |
|  | LLaVA1.5 7B (Longer Caption) | 5% | 82% | 13% | 15.59 |

## E. Reconstruction Experiments

### E.1. Architecture

We implement the nested VAE similar to the `AutoencoderKL` implementation in the `diffusers` library. The network configuration is:

- `sample_size`: 512

- `in_channels`: 3

- `out_channels`: 3

- `down_block_types`: [`DownEncoderBlock2D`] $\times$ 6

- `up_block_types`: [`UpDecoderBlock2D`] $\times$ 6

- `block_out_channels`: [64, 128, 256, 256, 512, 512]

- `layers_per_block`: 2

- `act_fn`: silu

- `latent_channels`: 4/8/16

- `norm_num_groups`: 32

- `mid_block_attention_head_dim`: 1

- `num_layers`: 8

The model sizes for different latent channels are shown below. For the discriminator, we use the pretrained StyleGAN (Karras et al., 2019).

| Nested VAE | $c = 4$ | $c = 8$ | $c = 16$ |
|---|---|---|---|
| # Params (M) | 187.45 | 187.50 | 187.61 |

### E.2. Training

We use $obj \in \{human\ faces, text\}$ for our perception-focused queries. We use the following training configuration:

- GPU: 64 NVIDIA A100

- Per-GPU batch size: 8

- Global batch size: 512

- Training steps: 1,000,000

- Optimizer: AdamW

  - `lr`: 0.0001
  - `beta1`: 0.9
  - `beta2`: 0.95
  - `weight_decay`: 0.1
  - `epsilon`: 1e-8
  - `gradient_clip`: 5.0

- Scheduler: constant with 10,000 warmup steps

- Loss:

  - `recon_loss_weight`: 1.0
  - `kl_loss_weight`: 1e-6
  - `perceptual_loss_weight`: 1.0
  - `moco_loss_weight`: 0.2
  - `gan_loss_weight`: 0.5
  - `gan_loss_starting_step`: 50,000

The discriminator is trained with the standard GAN loss.

### E.3. Baselines

We train fixed compression baselines using the same data, training configuration, and VAE backbone. For smaller compression ratios, e.g., fixed 8x, the last two downsampling blocks and first two upsampling blocks are not used.

For the adaptive JPEG baseline, we use torchvision.io.encode_jpeg to transform the images into JPEG file and compute the number of bytes as the complexity metric. Smaller files correspond to larger complexity. To provide a better understanding of this metric, we show in Figure 6 the distribution of JPEG sizes on the COCO 2014 test set, with relevant statistics included in the caption. Then, based on the JPEG sizes of all images in the Shutterstock training dataset, we set the thresholds $(a, b)$ to $(38761, 65837)$ to categorize the file sizes into three compression ratios. This set of thresholds ensure that the JPEG baseline has the same training compression ratio distribution as CAT.

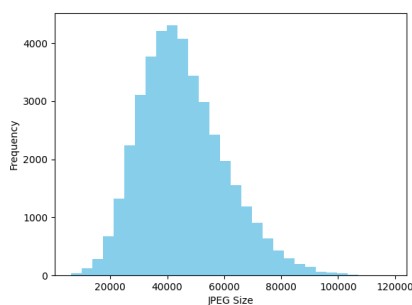

Figure 6: On COCO 2014 test set, the minimum JPEG size is 6128; maximum is 118428; mean is 45474.29; standard deviation is 15037.07.

For LDM VAEs, we follow the instructions in their original repository to use
the model checkpoints. Note that LDM VAEs are trained on OpenImages
dataset (Kuznetsova et al., 2020), which is different from our training data,
so it is hard to fairly compare the reconstruction performance. Nonetheless, we present their rFIDs on the evaluation datasets
in Table 11.

Table 11: rFIDs for CAT and LDM VAEs.

|         | COCO | ImageNet | CelebA | ChartQA |
|---------|------|----------|--------|---------|
| CAT     | 0.65 | 0.46     | 1.97   | 5.27    |
| LDM 8x  | 0.51 | 0.33     | 2.83   | 8.32    |
| LDM 16x | 0.53 | 0.37     | 3.07   | 8.49    |
| LDM 32x | 0.90 | 0.62     | 5.54   | 10.35   |

**E.4. More Reconstruction Visualization**

See Figure 7 in the next page.

### E.5. Fixing Token Count for CAT

We also evaluate the reconstruction performance under fixed compression ratio (token count) for different datasets. Table 12 compares the reconstruction FID for CAT with caption-guided compression ratio vs. fixed 16x compression ratio. We see that adaptive compression based on image complexity outperforms uniform compression using the same architecture in most cases, possibly because error reduction on complex images outweighs the slight error increase on simpler ones.

Table 12: Equalizing test-time token counts.

| CAT rFID | Adaptive | Equalized 16x |
| --- | --- | --- |
| COCO | **0.65** | 0.67 |
| ImageNet | 0.46 | **0.40** |
| CelebA | **1.97** | 2.47 |
| ChartQA | **5.27** | 7.27 |

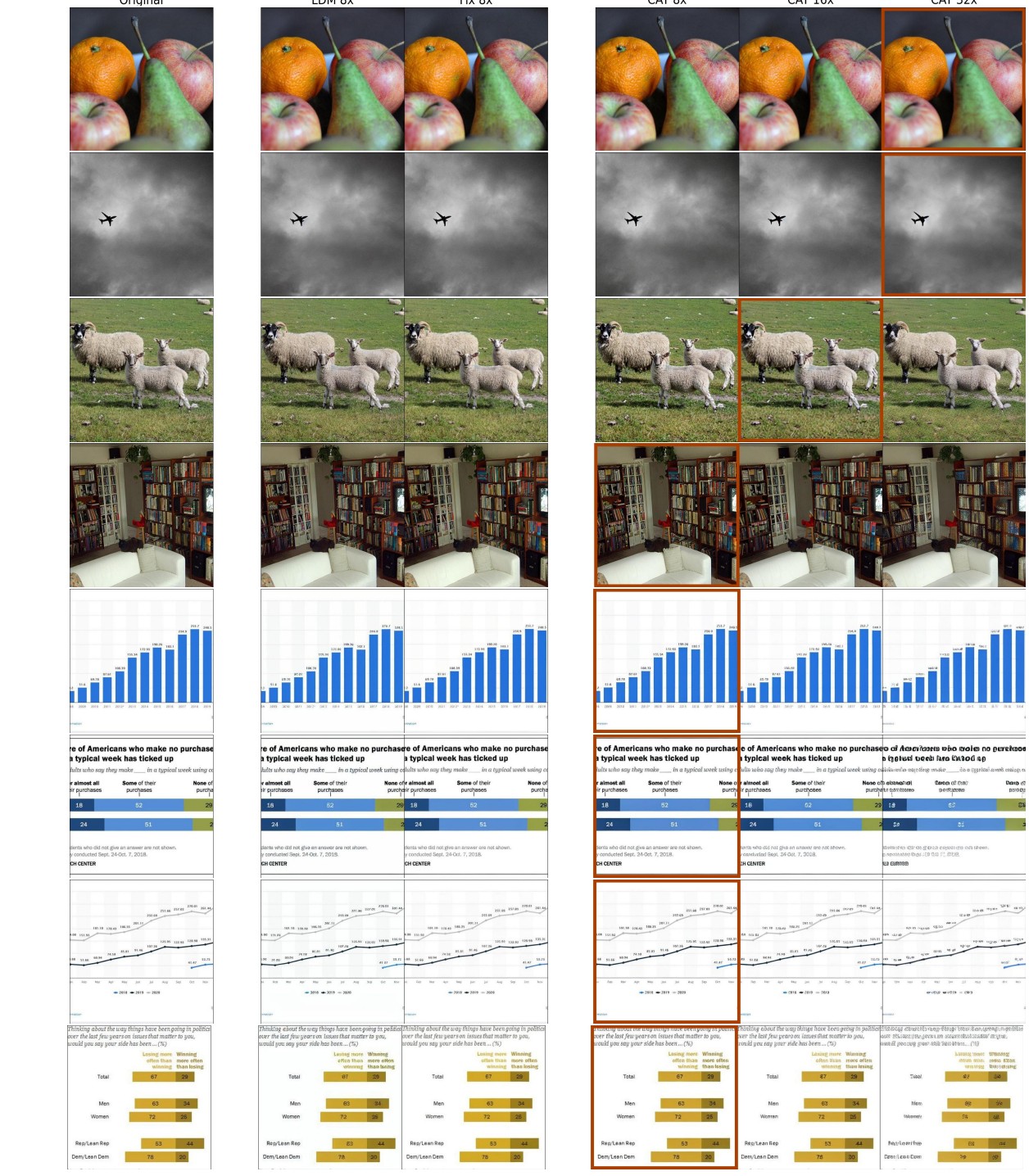

Figure 7: **More CAT reconstruction examples. We highlight the compression ratio selected by our caption complexity in red.**

### E.6. Full Results for Table 4 and Table 5

To complement Table 4, we include the training distribution for different scoring
thresholds in Table 13. To complement Table 5, we include the results of fixed-ratio baseline in Table 14.

Table 13: **Compression ratio distribution affects learning outcomes.** Both settings have an average compression of ~16x, but $(4, 7)$ leads to better distribution diversity and empirical results.

| $(a, b)$ | Training Distribution | | | | Reconstruction FID ↓ | | | |
|---|---|---|---|---|---|---|---|---|
| | 8x | 16x | 32x | Average | COCO | ImageNet | CelebA | ChartQA |
| $(4, 7)$ | 10% | 48% | 42% | 16.0x | **0.65** | 0.46 | **1.97** | **5.27** |
| $(2, 8)$ | 0.5% | 89.5% | 10% | 16.5x | 0.67 | **0.43** | 2.58 | 7.70 |

Table 14: **Larger latent channel $c$ generally improves rFID.**

| rFID↓ | $c$ | COCO | ImageNet | CelebA | ChartQA |
|---|---|---|---|---|---|
| | 4 | 1.25 | 1.32 | 5.89 | 9.45 |
| Fixed 16x | 8 | 1.10 | 0.61 | 4.99 | **8.19** |
| | 16 | **0.66** | **0.38** | **2.25** | 8.67 |
| | 4 | 1.66 | 1.10 | 5.83 | 9.13 |
| CAT | 8 | 1.03 | 0.60 | 4.54 | 7.95 |
| | 16 | **0.65** | **0.46** | **1.97** | **5.27** |

# F. Classification Experiments

We selected the diverse datasets because they represent out-of-domain distributions where zero-shot models perform poorly (Radford et al., 2021; Xu et al., 2024; Ilharco et al., 2022). Similar to OpenClip (Ilharco et al., 2021), we initialize a simple linear classification head that maps from the tokenizer's maximum latent dimension (under 8x compression) to the number of labels for each task. When the image is compressed with 16x or 32x ratio, we zero-pad the latent features to make the length match with the classification head. We evaluate two settings: (1) linear probing, where we keep the image encoder frozen and only fine-tune the classification head; (2) full fine-tuning, where we update both the encoder and the classification head. We train both settings for 20 epochs with a batch size of 64, learning rate 1e-4 and a cosine annealing learning rate schedule with 2 warm-up epochs. We use the AdamW optimizer with weight decay 0.1.

# G. Generation Experiments

## G.1. Architecture

We use DiT-XL architecture with a patchify downsampler and patch size of 2. The model size depends on the latent channel, but is generally around 431M parameters. The model TFLOPs is 22.0. All baselines reported in Table 7 use $c = 16$.

## G.2. Training & Inference

We use LLaVA1.5 7B to generate the complexity score for ImageNet training images. For 10 % of the time, we remove the image class label from the input and train unconditional image generation. The training configuration for DiT is:

- GPU: 16 NVIDIA H100

- Per-GPU token batch size: $4096 \times 4$ (equivalent to 64 images for 16x compression ratio)

- Global token batch size: $4096 \times 64$

- Training steps: 400,000

- Optimizer: AdamW

    - `lr`: 0.0001
    - `beta1`: 0.9
    - `beta2`: 0.95
    - `weight_decay`: 0.1
    - `epsilon`: 1e-8
    - `gradient_clip`: 1.0

- Scheduler: Cosine

    - `warmup`: 4000
    - `cosine_theta`: 1.0
    - `cycle_length`: 1.0
    - `lr_min_ratio`: 0.05

At test time, we use "*this is an image of [label]*" as a standardized prompt and manually provide answers to the queries to enable automated evaluation. Then, we use Llama3.1 to obtain the complexity score. DDPM scheduler (`diffusers` implementation) configuration is:

- `num_train_timesteps`: 1000

- `beta_start`: 0.0001

- `beta_end`: 0.02

- `beta_schedule`: squaredcos_cap_v2

- `prediction_type`: epsilon

- `timestep_spacing`: leading

- `num_inference_steps`: 250

All FID-50K and images generated in this paper are using cfg=1.5.

### G.3. Baselines

To ensure we train the baseline with the same compute FLOPs, we fix the token batch size and number of training steps for all settings. For pretrained LDM VAE, we use the scaling factor specified in the model configuration to ensure the input scale and noise scale are similar. For CAT, we use a scaling factor of 1.

### G.4. More Visualization

See Figure 8 in the end.

### G.5. Full Results for Table 8

To complement Table 8, we include the results of fixed-ratio baseline in Table 15.

Table 15: **Larger channel $c$ is not always better for generation.** Contrary to Table 5, we find that increasing channel dimension does not always result in generation gains.

|  | $c$ | FID↓ | sFID↓ | IS↑ | Precision↑ | Recall↑ |
|---|---|---|---|---|---|---|
| | 4 | 5.11 | 10.84 | 158.80 | 0.75 | 0.49 |
| Fixed 16x | 8 | 4.96 | **10.39** | **221.85** | **0.76** | **0.51** |
| | 16 | **4.78** | 11.81 | 187.47 | 0.72 | 0.49 |
| | 4 | 5.12 | 11.12 | 152.39 | 0.72 | 0.48 |
| CAT | 8 | **4.38** | **10.31** | 181.03 | **0.76** | 0.48 |
| | 16 | 4.56 | 10.55 | **191.09** | 0.75 | **0.49** |

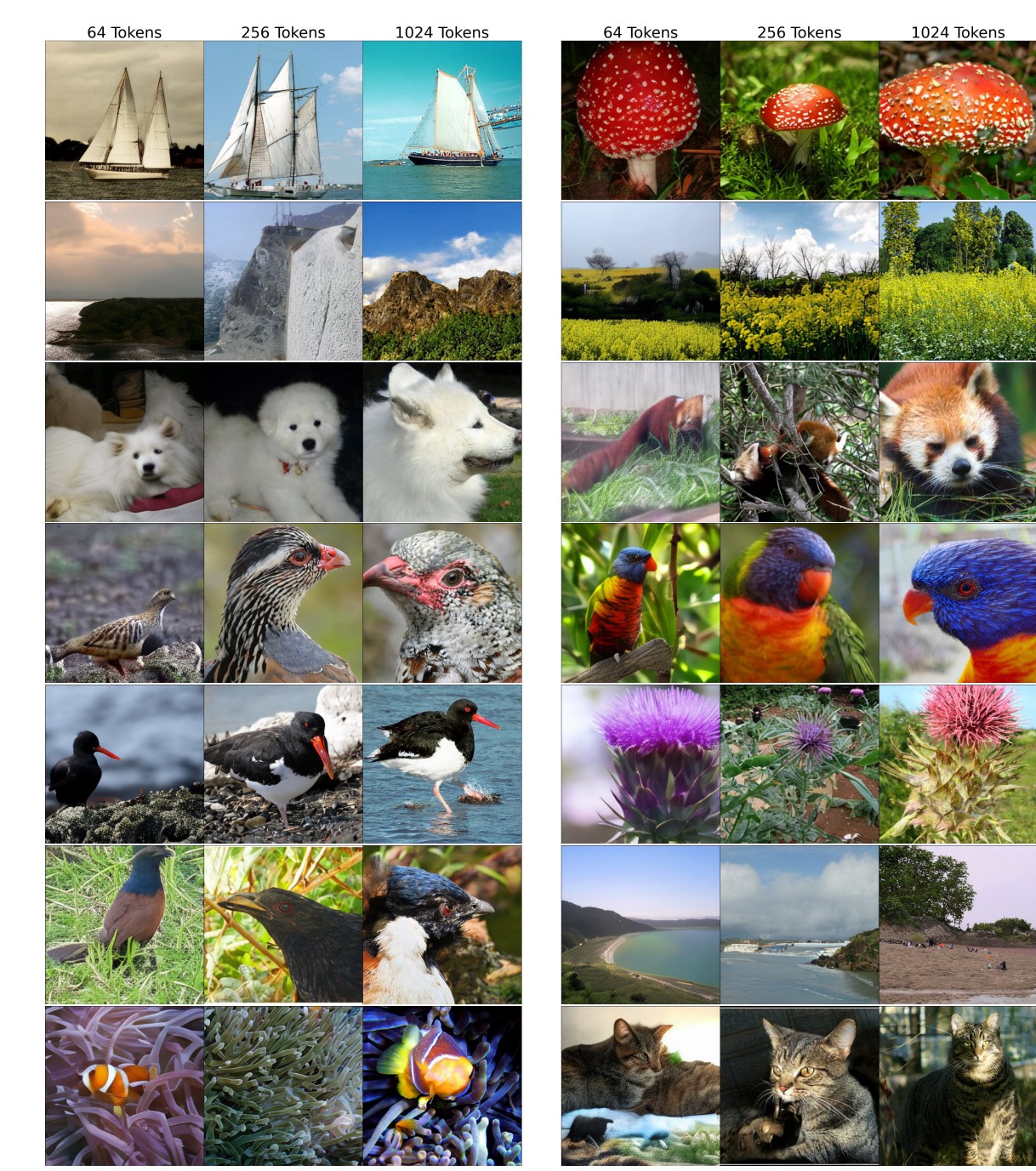

Figure 8: **More DiT-CAT generation examples. Increasing token count (left→right) generally leads to better image quality and higher complexity.**

