# OpenReview forum: "CAT: Content-Adaptive Image Tokenization"
_ICML.cc/2025/Workshop/TokShop — TokShop_

### Official Review · Reviewer_UmP3 · 2025-06-08
**Language-guided image tokenization. Relevant, interesting, and easy to follow.**

**Rating:** 7
**Confidence:** 3

**Review:**

The authors propose to have an LLM judge the complexity of an image; the complexity score then determines how much an image can be compressed based on a nested VAE (-8x, -16x, -32x). Some images with a low information content can be more heavily compressed compared to images with a lot of low-level details such as images containing text.

Strengths
- The paper is well-structured and easy to follow
- The authors explore and find robustness to different model choices and captioning variations.
- Meaningful ablation studies.
- The results have positive practical implications for other applications such as early-fusion VLMs

Weaknesses
- While maybe trivial to some, I would have appreciated a single sentence explaining the theme of the different datasets.
- The LLM-based complexity metric is central to the method. Consider including a deeper analysis into the limitation of this step. This could include model biases and ambiguity in the caption. Did the system systematically assign the "wrong" compression rate for certain types of images (mismatch between local and global complexity)?

---

### Official Review · Reviewer_9mcV · 2025-06-09
**Well structured paper and presents a novel approach around image tokenization**

**Rating:** 8
**Confidence:** 3

**Review:**

Summary:
This paper proposes a novel approach around image tokenization using LLMs to predict the image complexity and then suggest compression ratio and uses a nested VAE architecture for the image tokenization.

Strengths: This papers presents a novel approach. Overall idea and the way the paper has been structured is well articulated. The approach has been well tested around 4 datasets - COCO and ImageNet for natural images, CelebA and ChartQA
The results section clearly demonstrates the advantage of their approach.

---

### Decision · Program_Chairs · 2025-06-10

Accept